# Comparison of Multiple Models in Decentralized Target Estimation by a UAV Swarm

Fausto Francesco Lizzio [1], Martin Bugaj [2], Ján Rostáš [2] and Stefano Primatesta [1,*]

1 Department of Mechanical and Aerospace Engineering, Politecnico di Torino, Corso Duca degli Abruzzi 24, 10129 Torino, Italy; fausto.lizzio@polito.it
2 Air Transport Department, University of Žilina, Univerzitná 8215/1, 010 26 Žilina, Slovakia; martin.bugaj@uniza.sk (M.B.); jan.rostas@uniza.sk (J.R.)
* Correspondence: stefano.primatesta@polito.it

**Abstract:** The decentralized estimation and tracking of a mobile target performed by a group of unmanned aerial vehicles (UAVs) is studied in this work. A flocking protocol is used for maintaining a collision-free formation, while a decentralized extended Kalman filter in the information form is employed to provide an estimate of the target state. In the prediction step of the filter, we adopt and compare three different models for the target motion with increasing levels of complexity, namely, a constant velocity (CV), a constant turn (CT), and a full-state (FS) model. Software-in-the-loop (SITL) simulations are conducted in ROS/Gazebo to compare the performance of the three models. The coupling between the formation and estimation tasks is evaluated since the tracking task is affected by the outcome of the estimation process.

**Keywords:** decentralized target tracking; flocking; extended Kalman filter; consensus; multi-UAV

## 1. Introduction

In recent years, the distributed control of unmanned aerial vehicles (UAVs) has gained significant attention among researchers. Indeed, the scalability of modern control protocols, alongside the price convenience of the platforms, allows for more accessible planning of missions with multiple agents. This provides robustness against a single point of failure, increased coverage, and greater time efficiency.

Formation [1] is a prominent issue to tackle when controlling a group of UAVs, as drones commonly have to reach a safe relative distance during the execution of a mission. In the literature, several strategies are exploited to achieve this in a distributed way. Formation has been tackled through the use of artificial potential fields [2], treating each agent as a charged particle in an electric field. The geometric properties of the agents have been employed to deal with this problem [3]. Optimization methods [4–6] have also been explored to minimize a cost function subject to several constraints.

The aim of creating a formation usually revolves around a single [7] or multiple [8,9] targets to be chased by the swarm, to perform tasks such as bridge inspection, patrolling, or surveillance. Unlike many case studies reported in the literature [10,11], in real-life scenarios, the state of the target may not be accessible to the members of the swarm, as the target may be non-collaborative. In this situation, its precise state variables are unknown and must be estimated through certain estimation filters, as in [12]. Therefore, the agents have to follow the target location as provided by some collaborative estimation process, rather than by its exact position.

Hence, it is evident that the convergence of the estimation procedure is crucial for the outcome of the formation task [13]. The purpose of this paper is to outline and analyze an approach for a target estimation performed collaboratively onboard by a swarm of UAVs in formation flight.

A flocking consensus protocol, first established in [14], is adopted as a starting point for the formation task. In particular, we implement a tailored version of the algorithm in [14] presented by the authors in [15]. Indeed, the proposed version eliminates the errors occurring at steady-state in the relative distance between agents. Moreover, it is able to achieve smoother transient behavior with respect to the protocol in [14]. However, in [15], the target states were straightforwardly sent to all members of the swarm, so that the resulting topology behaved as a leader–followers one.

Therefore, in [16], the authors applied the formation protocol presented in [15] alongside a collaborative estimation method. In particular, the information form of a decentralized Kalman filter was used. The selection of this filter form was dictated by the many advantages it yields in decentralized sensor networks [17]. With this configuration, every drone acted as a mobile sensor whose measurements of the target state were affected by some noise. Namely, a range-bearing sensor was considered. The information provided by it was first employed locally and then integrated with the one coming from neighboring agents. This resulted in a leaderless topology that brought the configuration closer to an actual setup.

However, in [16], a linear process was selected to model the target dynamics during the prediction step of the filter. That is, the target was assumed to be a non-maneuvering one. When the estimation process was tested against a target covering a sinusoidal trajectory, the algorithm provided satisfactory results only when the interaction topology between UAVs was fully connected. Instead, when no information fusion was performed, the linearity of the process model was not able to handle the non-linear path, resulting in higher estimation errors.

That is why, in this work, we compare the performance of the linear process model, which we will refer to as the constant velocity (CV) model, with two non-linear models, i.e., a constant turn (CT) and a full-state (FS) model. The former assumes a constant turn rate for the target motion, while the latter takes into account the inner-loop dynamics of the target UAV. In this way, it is possible to evaluate the benefits of employing non-linear models, even though this leads to increased complexity and a higher computational load. Such an approach towards multiple model estimations has been applied in the literature for a single observer in [18,19], or for a team of agents in [20] to the classic Kalman filter formulation. Instead, this paper investigates the effects of different model estimations in the information form of the filter.

A robot operating system (ROS) is used to provide the simulation results. In particular, the software-in-the-loop (SITL) mode available through the PX4 autopilot stack [21] allows it to run multiple vehicles simultaneously, through the Gazebo simulator. Also, it is possible to exploit the navigation variables that would be accessible on-board actual UAVs, so that the autopilot can be interfaced as in a real implementation.

Metrics such as the estimation root mean square error and the drones' target distances will be presented to analyze the influence of the process model selection on the outcome of the process.

The remainder of this paper is organized as follows: Section 2 introduces the flocking protocol, alongside the tailored version developed by the authors. In Section 3, the decentralized estimation task is discussed and the difference between the three selected process models is highlighted. Section 4 presents the features of the simulation and illustrates the numerical examples. In Section 5, a general discussion of the results is given. Section 6 draws our conclusions and provides some comments on future work.

## 2. Tailored Flocking Algorithm

In this section, we discuss the formation protocol. In particular, we employ, as a starting point, the popular flocking algorithm established in [14]. In this paper, we denote it as the *standard protocol*. In [14], it is shown that this algorithm is able to provide, for double-integrator agents, the three flocking rules introduced in [22]. Namely, these rules

are: staying in the vicinity of the center of the swarm; avoiding collision with other agents; and having a common velocity value.

Consider a group of $N$ agents with double-integrator dynamics

$$\dot{\mathbf{q}}_i = \mathbf{p}_i, \quad \dot{\mathbf{p}}_i = \mathbf{u}_i, \quad \text{for } i = 1, ..., N,$$

where $\mathbf{q}_i = [x_i \ y_i]^T \in \mathbb{R}^2$ and $\mathbf{p}_i = [\dot{x}_i \ \dot{y}_i]^T \in \mathbb{R}^2$ indicate the position and velocity of agent $i$, respectively. The input of the *standard protocol* consists of the summation of three contributions, as

$$\mathbf{u}_i = \underbrace{\mathbf{u}_{i,d}}_{\text{distance regulator}} + \underbrace{\mathbf{u}_{i,v}}_{\text{velocity matching}} + \underbrace{\mathbf{u}_{i,t}}_{\text{target following}}. \tag{1}$$

The first one, $\mathbf{u}_{i,d}$, regulates the inter-agent distances to a desired value $d$. It is given by the gradient of a potential field acting between any two agents, whose minimum is located in $d$. The gradient is defined as

$$\mathbf{u}_{i,d} = -K_d \sum_{j \in \mathcal{N}_i} a_{ij} \cdot (\mathbf{q}_i - \mathbf{q}_j) \cdot \frac{\phi(\|\mathbf{q}_i - \mathbf{q}_j\|_\sigma - \|d\|_\sigma)}{1 + \epsilon \|\mathbf{q}_i - \mathbf{q}_j\|_\sigma},$$

where $K_d$ is a positive gain and $\mathcal{N}_i$ is the set of neighbors of agent $i$. The function $\phi(z)$ is defined as

$$\phi(z) = \frac{z}{\sqrt{1 + z^2}},$$

where $\| \cdot \|_\sigma$ indicates a map $\mathbb{R}^m \to \mathbb{R}_0^+$ differentiable everywhere given by $\|z\|_\sigma = \frac{1}{\epsilon}[\sqrt{1 + \epsilon \|z\|^2} - 1]$, with $\epsilon \in (0, 1)$. The parameter $a_{ij}$ depends on the communication range $r_{\text{comm}}$ and the relative distance between agents. In particular, $a_{ij} = a_{ij}(\mathbf{q}_i, \mathbf{q}_j) = a_{ij}(\frac{\|\mathbf{q}_i - \mathbf{q}_j\|_\sigma}{\|r_{\text{comm}}\|_\sigma})$ and

$$a_{ij}(z) = \begin{cases} 1, & \text{if } z \in [0, h) \\ \frac{1}{2}[1 + \cos(\pi \frac{z-h}{1-h})], & \text{if } z \in [h, 1] \\ 0, & \text{otherwise.} \end{cases}$$

In this way, the force repelling two agents that are dangerously close to each other is stronger than the attractive one. Furthermore, agents that are further than $r_{\text{comm}}$ do not contribute to the control input.

The second term $\mathbf{u}_{i,v}$ of Equation (1) is an average consensus step, expressed as

$$\mathbf{u}_{i,v} = -K_v \sum_{j \in \mathcal{N}_i} a_{ij} \cdot (\mathbf{p}_i - \mathbf{p}_j),$$

where $K_v$ is a positive gain. This term brings the agents to an agreement on their velocity values.

Finally, denoting $\mathbf{q}_t = [x_t \ y_t] \in \mathbb{R}^2$ and $\mathbf{p}_t = [\dot{x}_t \ \dot{y}_t] \in \mathbb{R}^2$ as the position and velocity of a target, the contribution

$$\mathbf{u}_{i,t} = -K_{d,t} \cdot (\mathbf{q}_i - \mathbf{q}_t) - K_{v,t} \cdot (\mathbf{p}_i - \mathbf{p}_t) \tag{2}$$

is a PD (proportional-derivative) controller driving the position and velocity of the agents toward the target ones, where $K_{d,t}$ and $K_{v,t}$ are positive gains.

However, the *standard protocol* presents some issues when it is applied to non-linear agents. First, a steady-state offset arises on the relative distance between agents, as found

in [23]. Hence, in [15], the authors modified the *standard protocol* to overcome this issue. Specifically, the error in the inter-agent distances

$$\mathbf{e}_{i,int} = \sum_{j \in \mathcal{N}_i} (\mathbf{q}_j - \mathbf{q}_i) \cdot \frac{\phi(\|\mathbf{q}_i - \mathbf{q}_j\|_\sigma - \|d\|_\sigma)}{1 + \epsilon\|\mathbf{q}_i - \mathbf{q}_j\|_\sigma},$$

allows us to define the following integral action

$$\mathbf{u}_{i,int} = K_{int} \int \mathbf{e}_{i,int} dt,$$

where $K_{int}$ is a positive gain. This term was included to eliminate the steady-state offset, even though the transient phase started to display a larger overshoot and considerable oscillations. Thus, the following integral action on the velocity mismatch between each agent and the target

$$\mathbf{u}_{i,v\_int} = -K_{v\_int} \int (\mathbf{p}_i - \mathbf{p}_t) dt$$

was further employed to dampen the initial transient, where $K_{v\_int}$ is a positive gain. Finally, a dynamic gain that multiplies the position error between the agents and the target was used, as

$$K_{d,t}(\mathbf{q}_i, \mathbf{q}_t) = \arctan(\frac{\|\mathbf{q}_i - \mathbf{q}_t\|}{D_{d,t}}). \tag{3}$$

This was done to decrease the attractive force of the target in its vicinity, where $D_{d,t} > 0$ is a parameter regulating the degree of this adjustment.

After these adaptations, the formation protocol could eliminate the steady-state offset while providing satisfactorily smooth transient behavior. A more thorough analysis of the flocking protocol can be found in a previous work by the authors [15]. In this study, the steps from the standard algorithm to the tailored one are showcased through numerical simulations.

## 3. Decentralized Target Estimation

The estimation task is discussed in this section. In particular, we make use of a decentralized variant of the extended Kalman filter in the information form, found in [17]. The Kalman filter is an iterative procedure designed to provide an estimate of the state variables of a dynamic process. A filter iteration is composed of a prediction step, employing a model of the process, and an update step, exploiting a sensor measurement. A centralized version of the Kalman filter provides the optimal estimate. The extended version of the Kalman filter is able to deal with non-linearities both in the process and in the sensor models. However, the need for a central processing unit and the collection of a large number of measurements hinder its scalability. This is why decentralized versions of the algorithm have emerged in the literature [24–26]. In this way, several agents can independently perform a measurement of the process and share some information to provide a cohesive estimate of the state variable. In this sense, the information form of the Kalman filter is of great advantage. Rather than the covariance matrix $\mathbf{P}(l|m)$ and the state estimate $\boldsymbol{\xi}(l|m)$ of a dynamic process, this formulation handles the information matrix $\boldsymbol{\Gamma}(l|m)$ and the information vector $\boldsymbol{\gamma}(l|m)$. These quantities are expressed as

$$\boldsymbol{\Gamma}(l|m) = \mathbf{P}^{-1}(l|m), \quad \boldsymbol{\gamma}(l|m) = \mathbf{P}^{-1}(l|m) \cdot \boldsymbol{\xi}(l|m), \tag{4}$$

where $l$ and $m$ are two generic time instants [27].

Although the formulation in [17] is algebraically equivalent to the classic filter, it presents appealing features when dealing with multi-sensor networks. Indeed, the dimension of the largest matrix to be inverted is linked to the state estimate one, rather than to the number of collective observations. In a UAV swarm, the number of measurements is generally much higher than the observed states. Furthermore, the information can be fused

through a trivial sum. Finally, assuming that no information about the process is known at the beginning of the estimation, the initialization procedure can be performed easily by assigning almost zero values to the information matrix. Given these considerations, we focus on the estimation of a target's position and velocity through an information filter, which will be discussed in the following subsections.

### 3.1. Prediction

The first step of an estimation filter is the prediction, in which the next state of the target is computed based on the previous estimate and on a selected dynamic model. Given the relevance of the choice of the model for the outcome of the estimation process [20], we employ three different models in the prediction step. In particular, we employ a constant velocity (CV), a constant turn (CT) and a full-state (FS) model, as described below. Note that, throughout the paper, all equations are expressed in their discrete-time formulation.

The CV model assumes a non-maneuvering target whose linear behavior is described by

$$\boldsymbol{\xi}_i(k+1) = \mathbf{F} \cdot \boldsymbol{\xi}_i(k) + \mathbf{w}(k), \tag{5}$$

where

$$\boldsymbol{\xi}_i(k) = [\hat{x}_{t,i}(k)\ \hat{y}_{t,i}(k)\ \hat{\dot{x}}_{t,i}(k)\ \hat{\dot{y}}_{t,i}(k)]^T \in \mathbb{R}^4$$

is the state estimate performed by agent $i$. Vector $\mathbf{w}(k) \in \mathbb{R}^4$ is an additive zero mean white noise whose covariance matrix is $\mathbf{Q}(k) \in \mathbb{R}^{4\times 4}$. The matrix $\mathbf{F}$ is simply defined as

$$\mathbf{F} = \begin{bmatrix} 1 & 0 & \Delta T & 0 \\ 0 & 1 & 0 & \Delta T \\ 0 & 0 & 1 & 0 \\ 0 & 0 & 0 & 1 \end{bmatrix},$$

where $\Delta T$ is the estimation time step. This is one of the simplest dynamic models found in the literature. It assumes that the target is either still, or moves along a straight line. It can provide mild performances in the case of low accelerations, but it is usually not suitable for highly non-linear trajectories, especially during sudden accelerations or sharp turns.

The CT model assumes a nearly constant turn rate $\omega_t$ of the target, so that its motion displays an almost zero tangential acceleration and a nearly constant normal acceleration [28]. If the turn rate is known, the model is linear and the corresponding prediction step is still given by Equation (5), with

$$\mathbf{F} = \begin{bmatrix} 1 & 0 & \frac{\sin(\omega_t \Delta T)}{\omega_t} & -\frac{1-\cos(\omega_t \Delta T)}{\omega_t} \\ 0 & 1 & \frac{1-\cos(\omega_t \Delta T)}{\omega_t} & \frac{\sin(\omega_t \Delta T)}{\omega_t} \\ 0 & 0 & \cos(\omega_t \Delta T) & -\sin(\omega_t \Delta T) \\ 0 & 0 & \sin(\omega_t \Delta T) & \cos(\omega_t \Delta T) \end{bmatrix},$$

as in [28]. However, considering a non-collaborative target, it is unlikely that the rate $\omega_t$ is known. Hence, its value must be estimated, so that the state estimate is augmented as

$$\boldsymbol{\xi}_i(k) = [\hat{x}_{t,i}(k)\ \hat{y}_{t,i}(k)\ \hat{\dot{x}}_{t,i}(k)\ \hat{\dot{y}}_{t,i}(k)\ \hat{\omega}_{t,i}]^T \in \mathbb{R}^5.$$

In this way, the process model becomes non-linear and the corresponding prediction equation, as in [20], is given by

$$\boldsymbol{\xi}_i(k+1) = \mathbf{f}(\boldsymbol{\xi}_i(k)) = \begin{pmatrix} \hat{x}_{t,i}(k) + \hat{\dot{x}}_{t,i}(k)\frac{\sin(\hat{\omega}_{t,i}(k)\Delta T)}{\hat{\omega}_{t,i}(k)} - \hat{\dot{y}}_{t,i}(k)\frac{1-\cos(\hat{\omega}_{t,i}(k)\Delta T)}{\hat{\omega}_{t,i}(k)} \\ \hat{y}_{t,i}(k) + \hat{\dot{x}}_{t,i}(k)\frac{1-\cos(\hat{\omega}_{t,i}(k)\Delta T)}{\hat{\omega}_{t,i}(k)} + \hat{\dot{y}}_{t,i}(k)\frac{\sin(\hat{\omega}_{t,i}(k)\Delta T)}{\hat{\omega}_{t,i}(k)} \\ \hat{\dot{x}}_{t,i}(k)\cos(\hat{\omega}_{t,i}(k)\Delta T) - \hat{\dot{y}}_{t,i}(k)\sin(\hat{\omega}_{t,i}(k)\Delta T) \\ \hat{\dot{x}}_{t,i}(k)\sin(\hat{\omega}_{t,i}(k)\Delta T) + \hat{\dot{y}}_{t,i}(k)\cos(\hat{\omega}_{t,i}(k)\Delta T) \\ \hat{\omega}_{t,i}(k) \end{pmatrix} + \mathbf{w}(k),$$

where $\mathbf{w}(k) \in \mathbb{R}^5$ is an additive zero mean white noise. This model works better than the CV one when dealing with curvilinear motions, especially when the tangential acceleration is very small with respect to the normal one. It can also provide a mild performance when the target moves along linear trajectories, as an almost zero turn rate can be assumed. However, its performance decreases when the tangential acceleration is comparable to the normal one.

Finally, the FS model takes into account the inner loop dynamics of a quad-copter UAV, considering the attitude angles and rates of the target along with its planar positions and velocities, as in [29]. Moreover, in dealing with a non-collaborative target, the state estimate has to include the unknown target control inputs. Since the analyzed motion is planar, we do not consider the information regarding the altitude coordinate. Also, in the target following problem, we treat the target as a point, so that its yaw-related states are disregarded. Then, we obtain

$$\boldsymbol{\zeta}_i(k) = [\hat{x}_{t,i}(k) \; \hat{y}_{t,i}(k) \; \hat{\dot{x}}_{t,i}(k) \; \hat{\dot{y}}_{t,i}(k) \; \hat{\dot{z}}_{t,i}(k) \; \hat{\phi}_{t,i}(k) \; \hat{\theta}_{t,i}(k) \; \hat{\dot{\phi}}_{t,i}(k) \; \hat{\dot{\theta}}_{t,i}(k) \; \hat{U}_{\phi,i}(k) \; \hat{U}_{\theta,i}(k) \; \hat{T}_i(k) \;]^T \in \mathbb{R}^{12}.$$

In addition to the planar position and velocity, the state estimate contains the roll and pitch angles $\hat{\phi}_{t,i}(k)$ and $\hat{\theta}_{t,i}(k)$, their derivatives $\hat{\dot{\phi}}_{t,i}(k)$ and $\hat{\dot{\theta}}_{t,i}(k)$, the rolling and pitching moments $\hat{U}_{\phi,i}(k)$ and $\hat{U}_{\theta,i}(k)$, and the total thrust $\hat{T}_i(k)$. The derivative of the altitude coordinate $\hat{\dot{z}}_{t,i}(k)$ is needed to make the thrust converge to a reasonable value, as explained later. The resulting prediction equations are non-linear, and are given by

$$\boldsymbol{\zeta}_i(k+1) = \mathbf{f}(\boldsymbol{\zeta}_i(k)) = \begin{bmatrix} \hat{x}_{t,i}(k) + \Delta T \hat{\dot{x}}_{t,i}(k) \\ \hat{y}_{t,i}(k) + \Delta T \hat{\dot{y}}_{t,i}(k) \\ \hat{\dot{x}}_{t,i}(k) + \Delta T \frac{\hat{T}_i(k)}{m} sin(\hat{\theta}(k)) \\ \hat{\dot{y}}_{t,i}(k) - \Delta T \frac{\hat{T}_i(k)}{m} sin(\hat{\phi}(k)) cos(\hat{\theta}(k)) \\ -g + cos(\hat{\phi}(k)) cos(\hat{\theta}(k)) \frac{\hat{T}_i(k)}{m} \\ \hat{\phi}_{t,i}(k) + \Delta T \hat{\dot{\phi}}(k) \\ \hat{\theta}_{t,i}(k) + \Delta T \hat{\dot{\theta}}(k) \\ \hat{\dot{\phi}}(k) + \Delta T \hat{U}_{\phi,i}(k) \frac{l}{I_{xx}} \\ \hat{\dot{\theta}}(k) + \Delta T \hat{U}_{\theta,i}(k) \frac{l}{I_{yy}} \\ \hat{U}_{\phi,i}(k) \\ \hat{U}_{\theta,i}(k) \\ \hat{T}_i(k) \end{bmatrix} + \mathbf{w}(k),$$

where $\mathbf{w}(k) \in \mathbb{R}^{12}$ is an additive zero mean white noise. The parameters $m$, $l$, $I_{xx}$, and $I_{yy}$ are the mass, the arm length, and the moments of inertia along the $x$- and $y$-axis, respectively. Thus, the implementation of this model requires the knowledge of the physical parameters mentioned of the target UAV. As briefly mentioned, the altitude coordinate derivative is present in the state estimate, despite our motion of interest being planar. The term $\hat{\dot{z}}_{t,i}(k)$ and its related dynamics are required to make the filter aware that a certain thrust level is needed to keep the target flying at a fixed altitude. A way to force this behavior into the filter is to augment the measurement model through a fake altitude rate measurement constantly equal to zero.

Given the previous considerations, in the case of a linear process model, the prediction step of the filter is expressed as

$$\begin{cases} \boldsymbol{\Gamma}_i(k|k-1) = (\mathbf{F} \cdot \boldsymbol{\Gamma}_i^{-1}(k-1|k-1) \cdot \mathbf{F}^T + \mathbf{Q}(k))^{-1} \\ \boldsymbol{\gamma}_i(k|k-1) = \boldsymbol{\Gamma}_i(k|k-1) \cdot \mathbf{F} \cdot \boldsymbol{\Gamma}_i^{-1}(k-1|k-1) \cdot \boldsymbol{\gamma}_i(k-1|k-1). \end{cases}$$

When a non-linear dynamic model is selected to represent the target motion, the following prediction step must be adopted

$$
\begin{cases}
\boldsymbol{\Gamma}_i(k|k-1) = (\nabla \mathbf{f}(\boldsymbol{\xi}_i(k-1|k-1)) \cdot \boldsymbol{\Gamma}_i^{-1}(k-1|k-1) \cdot \nabla \mathbf{f}^T(\boldsymbol{\xi}_i(k-1|k-1)) + \mathbf{Q}(k))^{-1} \\
\boldsymbol{\gamma}_i(k|k-1) = \boldsymbol{\Gamma}_i(k|k-1)\mathbf{f}(\boldsymbol{\xi}_i(k-1|k-1)),
\end{cases}
$$

where $\nabla \mathbf{f}$ represents the gradient operator. In both cases, the variables $\boldsymbol{\Gamma}_i(k-1|k-1)$ and $\boldsymbol{\gamma}_i(k-1|k-1)$ can have values close to zero at the beginning of the task, indicating that there is almost no knowledge of the initial state of the process. This kind of initialization avoids the need to guess the actual initial target state.

### 3.2. Observation

After the prediction step is performed, the filter needs to employ the sensors' measurements. The observations are taken by $i = 1, \ldots, N$ agents whose measurement model is given by

$$
\mathbf{z}_i(k) = \mathbf{h}_i(\mathbf{q}_t(k)) + \mathbf{v}_i(k),
$$

with

$$
\mathbf{h}_i(\mathbf{q}_t(k)) = \begin{pmatrix} \sqrt{(x_t(k) - x_i(k))^2 + (y_t(k) - y_i(k))^2} \\ \arctan(\frac{y_t(k) - y_i(k)}{x_t(k) - x_i(k)}) \end{pmatrix},
$$

i.e., a range-bearing sensor affected by an additive zero mean white noise $\mathbf{v}_i(k) \in \mathbb{R}^2$ whose covariance matrix is $\mathbf{R}_i(k) \in \mathbb{R}^{2 \times 2}$.

The measurement collected by sensor $i$ allows to locally compute

$$
\begin{cases}
\mathbf{I}_i(k) &= \nabla \mathbf{h}_i^T(\hat{\mathbf{q}}_{t,i}(k|k-1)) \cdot \mathbf{R}_i^{-1}(k) \cdot \nabla \mathbf{h}_i(\hat{\mathbf{q}}_{t,i}(k|k-1)) \\
\mathbf{i}_i(k) &= \nabla \mathbf{h}_i^T(\hat{\mathbf{q}}_{t,i}(k|k-1)) \cdot \mathbf{R}_i^{-1}(k) \cdot (\mathbf{z}_i(k) - \mathbf{h}_i(\hat{\mathbf{q}}_{t,i}(k|k-1)) + \nabla \mathbf{h}_i(\hat{\mathbf{q}}_{t,i}(k|k-1))\boldsymbol{\xi}_i(k|k-1))
\end{cases} \tag{6}
$$

where $\hat{\mathbf{q}}_{t,i}(k|k-1)$ indicates the prediction of the target position performed by UAV $i$. The matrix $\mathbf{I}_i(k)$ and vector $\mathbf{i}_i(k)$ denote the information obtained by the agent $i$ after a measurement is taken.

### 3.3. Update

Lastly, the update step is expressed as

$$
\begin{cases}
\boldsymbol{\Gamma}_i(k|k) &= \boldsymbol{\Gamma}_i(k|k-1) + \mathbf{I}_i(k) + \sum_{j \in \mathcal{N}_i} \mathbf{I}_j(k) \\
\boldsymbol{\gamma}_i(k|k) &= \boldsymbol{\gamma}_i(k|k-1) + \mathbf{i}_i(k) + \sum_{j \in \mathcal{N}_i} \mathbf{i}_j(k),
\end{cases} \tag{7}
$$

that is interpreted as the sum of three contributions. The first contribution comes from the prediction step. The second one arises from agent $i$'s own measurement, while the last contribution comes from the measurements performed by all of its neighbors. This step clearly shows how convenient information fusion is with this filter formulation.

Through $\boldsymbol{\Gamma}_i(k|k)$ and $\boldsymbol{\gamma}_i(k|k)$ and the inverse of Equation (4), each agent can compute a local estimate $\boldsymbol{\xi}_i(k|k)$ of the target state. Then, the estimated planar positions and velocities of the target can be employed by the target following term in Equation (2) of the flocking protocol. By doing so, each agent tracks the estimated state of the target, rather than the true one. According to [12], the estimation of this decentralized Kalman filter is capable of producing the same estimates locally that a centralized filter would give, provided that all agents communicate through a fully connected topology.

## 4. Results

In this section, we present numerical results to illustrate the described application and compare the proposed dynamic models.

The flocking algorithm and the decentralized estimation using the three different models are implemented in C++ exploiting the robot operating system (ROS). Thus, the Gazebo simulator and the SITL mode available through the PX4 autopilot are used to

carry out simulation experiments. Specifically, we simulate three Iris quadcopters through three separate ROS nodes. Through the publisher/subscriber communication pattern, the nodes share with each other the information described in the previous sections. Then, the computed control input is published via Mavros to the PX4 Autopilot in the offboard mode. The UAVs have to attain a flocking task whose desired inter-agent distance is $d = 4\,\text{m}$ and whose communication radius is $r_{\text{comm}} = 4.8\,\text{m}$. The previously discussed EKF is employed to perform a decentralized estimate of the state of a target Iris platform. A global reference system has to be shared among all the members of the swarm to yield coherent information fusion.

To evaluate the performance of the three models, the target trajectory is split into three phases. First, the target moves in a straight line, with a constant velocity of $\dot{x}_t(t) = 0.5\,\text{m/s}$ along the $x$-axis, and a null velocity on the $y$-axis. We refer to this phase as the *linear* one. Second, the target follows a *sinusoidal* path with a velocity on the $y$-axis equal to $\dot{y}_t(t) = 2\sin(0.1t)\,\text{m/s}$, while keeping the previous constant velocity $\dot{x}_t(t) = 0.5\,\text{m/s}$ on the $x$-axis. Finally, in the third phase, the target switches to a *circular* motion at a higher frequency, while still proceeding along the $x$-axis. The velocity values in this phase are $\dot{x}_t(t) = (0.2 + 2\sin(0.25t))\text{m}\,/\,\text{s}$ and $\dot{y}_t(t) = 2\cos(0.25t)\,\text{m/s}$. The different types of trajectories are chosen to challenge the capabilities of the selected models.

We assume that the process noise covariance matrix is constant, so that for the CV model

$$\mathbf{Q} = \text{diag}\Big(0.05^2, 0.05^2, 0.05^2, 0.05^2\Big),$$

for the CT model

$$\mathbf{Q} = \text{diag}\Big(0.05^2, 0.05^2, 0.05^2, 0.05^2, 0.02^2\Big),$$

and for the FS model

$$\mathbf{Q} = \text{diag}\Big(0.05^2, 0.05^2, 0.05^2, 0.05^2, 0.05^2, 0.02^2, 0.02^2, 0.02^2, 0.02^2, 0.01^2, 0.01^2, 0.01^2\Big).$$

Instead, we assume that the measurement noise covariance matrix is expressed as

$$\mathbf{R}_i(k) = \Lambda_i^T \begin{bmatrix} 0.08^2 & 0 \\ 0 & \rho_i^2 0.02^2 \end{bmatrix} \Lambda_i$$

where $\rho_i = \sqrt{(\hat{x}_{t,i}(k|k-1) - x_i(k))^2 + (\hat{y}_{t,i}(k|k-1) - y_i(k))^2}$ is the estimated distance between UAV $i$ and the target, and

$$\Lambda_i = \begin{bmatrix} \cos(\hat{\beta}_i) & -\sin(\hat{\beta}_i) \\ \sin(\hat{\beta}_i) & \cos(\hat{\beta}_i) \end{bmatrix}$$

is a rotation matrix with $\hat{\beta}_i = \arctan(\frac{\hat{y}_{t,i}(k|k-1) - y_i(k)}{\hat{x}_{t,i}(k|k-1) - x_i(k)})$ being the estimated relative bearing angle, as in [30]. The term $\rho_i$ in $\mathbf{R}_i(k)$ makes the covariance matrix coefficients shrink as the relative distance between the agent and the target decreases. Hence, sensors provide measurements with lower covariance matrices as they get closer to the target location [13].

The estimation task is performed at 20 Hz. Since no a priori information is assumed to be known at the beginning of the mission, the initial estimate values are far from the true target states, and the estimation process requires several observation steps before converging to reasonable values. Thus, the drones only begin to chase the target after the first 20 iterations, i.e., after 1 second, so that the initial estimates do not drift them away.

In this study, we carried out three rounds of simulations, one for each dynamic model described previously. This was done to assess the advantages of using a more complex dynamic model in the prediction step. Note that increasing the level of complexity of the prediction model in an information filter comes with the price of inverting higher dimension matrices. In our simulation scenario, each agent must perform the inversion of a $4 \times 4$ matrix for the CV model, of a $5 \times 5$ matrix for the CT model, and of a $12 \times 12$ matrix for the FS

model. Instead, a classic Kalman filter would need each agent to invert a $6 \times 6$ matrix, given by the total number of observations (two measurements performed by three UAVs). Thus, it is straightforward to notice that, in terms of computational complexity, it is convenient to use the FS model only when the number of UAVs in the swarm is sufficiently large. Similar reasoning can be applied with respect to the communication cost due to the filter decentralization. Indeed, denote as $n$ the dimension of the state estimate, and as $|\mathcal{N}_i|$ the number of neighbors of agent $i$. Thus, at each filter iteration, agent $i$ sends out a message of size $n(n + 1)$ and collects $|\mathcal{N}_i|$ messages of the same size. However, in this work, the computational and communication costs are not the main issues, as the focus is establishing how the performance of the filter changes with different dynamic models.

In all scenarios, each drone shares the measurement information computed in Equation (6) according to the line interaction topology shown in Figure 1.

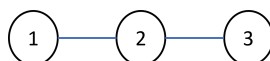

**Figure 1.** UAVs Interaction Topology.

In this way, only agent 2 can access the complete information contained in the graph. This represents a middle way between a fully connected topology and a situation in which no communication takes place. Refer to a previous work by the authors in [16] for numerical simulations showcasing the effects of these two opposite configurations. In the first case, all the information is available to each agent, so that the swarm mimics a centralized filter, but a higher number of communication links is demanded. In the latter case, each agent carries on the estimation process in an isolated manner. Thus, a line topology represents a compromise between performance and communication cost.

The estimates provided by the three UAVs of the target positions and velocities on the $x$- and $y$-axis are shown in Figure 2 for the CV, in Figure 3 for the CT, and in Figure 4 for the FS.

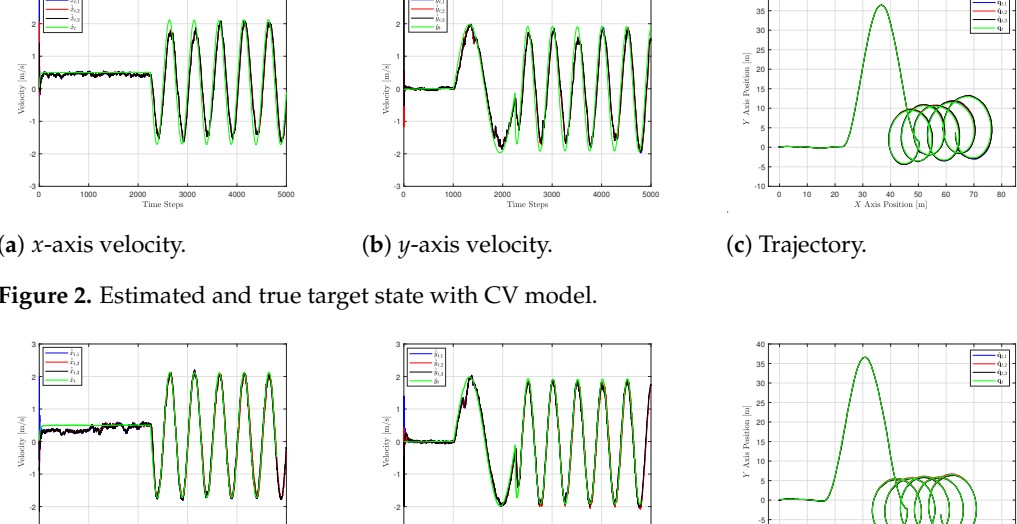

(**a**) $x$-axis velocity.  (**b**) $y$-axis velocity.  (**c**) Trajectory.

**Figure 2.** Estimated and true target state with CV model.

(**a**) $x$-axis velocity.  (**b**) $y$-axis velocity.  (**c**) Trajectory.

**Figure 3.** Estimated and true target state with CT model.

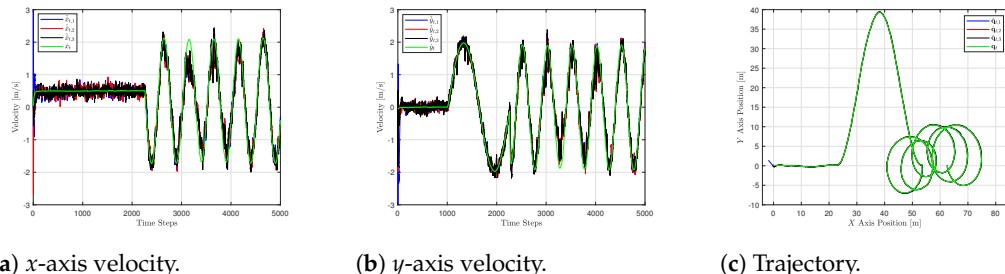

(**a**) *x*-axis velocity.          (**b**) *y*-axis velocity.          (**c**) Trajectory.

**Figure 4.** Estimated and true target state with FS model.

In particular, we report the true target states $\mathbf{q}_t(t)$ and $\mathbf{p}_t(t)$ in green, as well as the estimates $\hat{\mathbf{q}}_{t,i}(k|k)$ and $\hat{\mathbf{p}}_{t,i}(k|k)$ provided by the UAVs $i = 1, 2, 3$. Note that the simulation of the target UAV runs in a ROS node through the offboard mode, as described at the beginning of the section for the three drones in the swarm. Hence, its trajectory is not ideal and is affected by realistic noises and localization errors. This is why, in Figures 2–4, the trajectories of the target do not appear to be exactly the same.

We also report, in Tables 1–3, the mean of the root mean square errors (RMSEs) over the three UAVs for each model. It is clear that the performance of the estimation filter is affected by the choice of the model.

**Table 1.** Estimates RMSE of CV model.

|  | $\sigma_{\hat{x}}$ [cm] | $\sigma_{\hat{y}}$ [cm] | $\sigma_{\hat{\dot{x}}}$ [cm/s] | $\sigma_{\hat{\dot{y}}}$ [cm/s] |
|---|---|---|---|---|
| Linear | 8.30 | 1.96 | 7.59 | 3.02 |
| Sinusoidal | 6.77 | 20.55 | 7.28 | 23.40 |
| Circular | 27.11 | 27.71 | 47.40 | 48.21 |
| Total | 20.77 | 23.25 | 35.54 | 37.67 |

**Table 2.** Estimates RMSE of CT model.

|  | $\sigma_{\hat{x}}$ [cm] | $\sigma_{\hat{y}}$ [cm] | $\sigma_{\hat{\dot{x}}}$ [cm/s] | $\sigma_{\hat{\dot{y}}}$ [cm/s] |
|---|---|---|---|---|
| Linear | 6.47 | 2.12 | 18.51 | 4.29 |
| Sinusoidal | 8.18 | 23.44 | 8.51 | 20.94 |
| Circular | 26.76 | 25.80 | 11.32 | 15.18 |
| Total | 20.53 | 22.54 | 12.59 | 15.52 |

**Table 3.** Estimates RMSE of FS model.

|  | $\sigma_{\hat{x}}$ [cm] | $\sigma_{\hat{y}}$ [cm] | $\sigma_{\hat{\dot{x}}}$ [cm/s] | $\sigma_{\hat{\dot{y}}}$ [cm/s] |
|---|---|---|---|---|
| Linear | 6.48 | 4.68 | 13.80 | 15.37 |
| Sinusoidal | 6.26 | 14.85 | 10.73 | 12.17 |
| Circular | 18.24 | 19.73 | 33.95 | 34.02 |
| Total | 14.22 | 16.70 | 26.49 | 27.04 |

The CV handles linear motion very well. Its related position and velocity estimates provide low RMSE values, as seen in Table 1. Moving from the linear to the sinusoidal phase, the RMSEs regarding the *X* coordinate remain comparable, due to the fact that the velocity along this axis is still constant. However, it is possible to notice a substantial increase in the RMSE provided by the estimates in the *Y* coordinates. Finally, the strong non-linearity of the circular maneuver makes the CV model performance drop drastically, especially in the velocity estimation. This shows the poor performance of the model when dealing with highly maneuvered trajectories.

Regarding the CT model, the initial linear path challenges its capability in tracking the constant non-zero velocity along the *x*-axis. Indeed, while the position estimates are still

satisfactorily close to the real ones, the *X* velocity estimate is quite far from the actual value in this phase. The CT performs similarly to the CV model during the sinusoidal phase. Indeed, despite being curvilinear, this motion is not characterized by a constant turn rate. Instead, the position and velocity estimates during the circular phase yield lower estimation errors. A remarkable improvement can be noticed, especially in the velocity estimates RMSE. As seen in Table 2, they are about four times smaller than the ones provided by the CV model.

Lastly, the FS provides, in general, noisier estimates with respect to the previous two models, as seen in Figure 4. However, the analysis of its RMSEs in Table 3 provides some insights. The FS handles the first linear trajectory with average performances with respect to the other two models, especially from the velocity RMSE point of view. However, it performs quite well in the sinusoidal phase, providing lower estimation errors with respect to the CV and CT models. Finally, in the circular motion, it yields again average RMSEs with respect to the two other models. This may be due to the fact that, while the CV and the CT are specifically designed for a linear and a circular path, the FS does not assume anything about the target's motion. Thus, it behaves worse than these models in the first and last phases, while it outperforms them in the second one.

The different performances of the estimation process influence the flocking behavior. Indeed, the two tasks are executed in cascade, and the target tracking term in Equation (2) follows the estimates provided by the filter. In Figures 5–7, it is possible to analyze this influence. In particular, in Figure 5, the actual trajectories of the target and the agents are presented. For clarity of the pictures, only agent 1 is considered. Figure 6 displays the relative distances between the target and the three UAVs in the swarm, while Figure 7 showcases the inter-agent distances between the three agents, for each described model. Looking at the trajectory of agent 1 in Figure 5, it is possible to see that, overall, the three models can make the UAVs correctly and timely follow the target. Some more in-depth differences may be noticed in Figures 6 and 7. During the linear phase, the three models provide more or less the same flocking behavior. The desired inter-agent distance of 4 m is reached in all cases with a satisfactory transient and is kept almost constant. However, when the sinusoidal part takes place, an oscillatory behavior emerges. This is because, in the proximity of the target, the flocking protocol has an adaptive gain as in Equation (3), so that the reduced attractive force makes the agents less prompt in following the target. However, this distance is kept reasonably small and this behavior may be adjusted through parameter $D_{d,t}$ for a prompter formation. The FS provides the smoothest transient, while the CV and CT models act similarly. This can be clearly seen in Figure 7, where the inter-agent distances remain almost constant also in the sinusoidal phase. Finally, when the circular motion starts, the CV model yields the highest oscillations both in Figures 6 and 7. Instead, the CT provides lower oscillations, especially in terms of the target–agent distances. Finally, the FS model results in the lowest oscillations in both metrics. The better performance of the FS with respect to the CT is due to the noisier nature of the velocity estimates that it provides. Indeed, the actual velocity tracked by the agents through (2) ends up being smaller with the FS compared to the CT, resulting in lower oscillations.

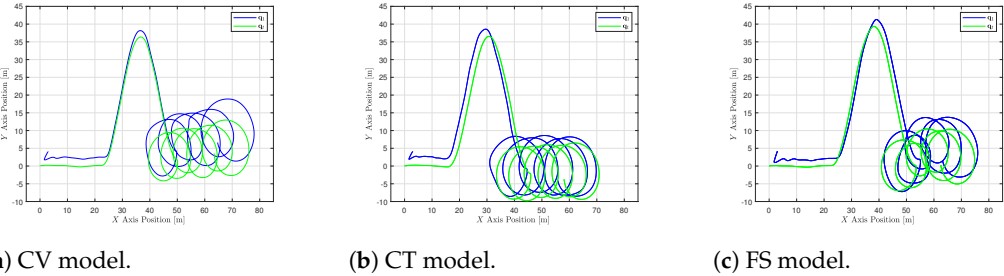

(**a**) CV model.  (**b**) CT model.  (**c**) FS model.

**Figure 5.** Target and agent 1 trajectories.

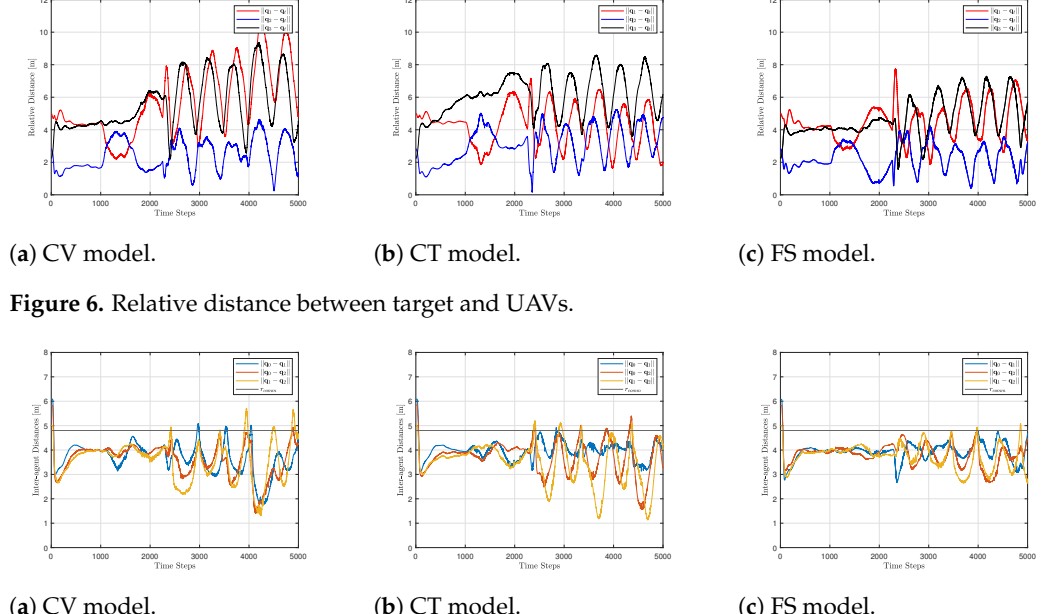

(**a**) CV model.　　　　　　(**b**) CT model.　　　　　　(**c**) FS model.

**Figure 6.** Relative distance between target and UAVs.

(**a**) CV model.　　　　　　(**b**) CT model.　　　　　　(**c**) FS model.

**Figure 7.** Inter-agent distances and communication range.

## 5. Discussion

The previous examples showed the performance of the EKFs with three different process models. It is clear that, in the presence of a non-maneuvering target, the CV model is the best choice, given its computational simplicity. However, its performance starts to degrade and eventually provides large RMSE with highly non-linear trajectories. The CT model represents a good choice when dealing with maneuvering targets, especially considering the very slight increase in computational complexity with respect to the CV model. Analyzing the RMSEs of the total trajectory in Tables 1–3, the FS appeared to be a middle way between the two previously mentioned models. This may be due to the fact that the FS model makes no assumption on any specific target motion, but rather it relies on the UAV's intrinsic characteristics. Also from the flocking and target tracking application point of view, the FS provided average performance with respect to the other two models, as seen in Figures 6 and 7. However, its computational complexity is considerably higher with respect to the CV and CT, since the information filter on board every UAV has to invert a $12 \times 12$ matrix. Moreover, it requires the knowledge of some physical properties of the target UAV, such as its mass, moments of inertia, and arm length. This may not always be the case in real missions, so these parameters should be added to the estimation vector, thus further increasing the computational load. Hence, employing the FS model would be convenient only when a very large swarm tracks a maneuvered UAV whose physical parameters are known but whose motion is generic and can not be clearly categorized either as CV or as CT.

Overall, the study shows that decentralized target tracking is feasible and can be implemented with satisfactory results even using a simple model and a topology that is not fully connected. This may be useful in the setup of a configuration for experimental tests. Indeed, one may think of using a CT model for preliminary results on a static target, as an Aruco marker on the ground. Then, the model complexity may be increased in further tests with moving targets. From a communication point of view, a line topology could be assigned beforehand in preliminary tests. Then, some more complex topologies can be implemented to increase the performance of the estimation algorithm. Additionally, an actual proximity graph may be adopted, so that agents only share their information with closest neighbors. This could bring the experiments much closer to a real case study.

## 6. Conclusions

In this paper, a decentralized version of the Kalman filter in the information form was employed to generate a collaborative estimate of a target state by a UAV swarm. The target moved along different kinds of paths, namely a straight, a sinusoidal, and a circular one. Three models were adopted in the prediction step of the filter, and their performance was compared in simulations using the ROS/Gazebo SITL framework. The numerical results showed that the model selection influences the outcome of the estimation and tracking tasks. As a next step, the models will be expanded to handle a target moving in a three-dimensional environment, so that the altitude coordinate will also be considered through the additional measurement of the relative elevation angle. Future research directions will also explore the use of interacting models so that an agent can switch between different process models while the estimation task is being executed. Moreover, the possibility of merging the benefits of the multiple models through a consensus step in the filter will be explored.

**Author Contributions:** Conceptualization, F.F.L. and S.P.; methodology, F.F.L. and S.P.; software, F.F.L.; validation, F.F.L.; investigation, F.F.L. and S.P.; data curation, F.F.L.; writing—original draft preparation, F.F.L.; writing—review and editing, F.F.L., S.P. and M.B.; visualization, F.F.L. and S.P.; supervision, S.P.; project administration, S.P., J.R. and M.B.; funding acquisition, S.P., J.R. and M.B. All authors have read and agreed to the published version of the manuscript.

**Funding:** This research received no external funding.

**Institutional Review Board Statement:** Not applicable.

**Informed Consent Statement:** Not applicable.

**Data Availability Statement:** The data used in the current study are available from the corresponding author upon reasonable request.

**Acknowledgments:** This paper is an output of the project 313011ATR9 Research and development of the usability of autonomous UAVs in the fight against the pandemic caused by COVID-19.

**Conflicts of Interest:** The authors declare no conflicts of interest.

## Abbreviations

The following abbreviations are used in this manuscript:

| | |
|---|---|
| CT | Constant Turn |
| CV | Constant Velocity |
| EKF | Extended Kalman Filter |
| FS | Full State |
| PD | Proportional-Derivative |
| RMSE | Root Mean Square Error |
| ROS | Robot Operating System |
| SITL | Software In The Loop |
| UAV | Unmanned Aerial Vehicle |

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
