# Peer review of "Comparison of Multiple Models in Decentralized Target Estimation by a UAV Swarm"

_drones, doi:10.3390/drones8010005_

Round 1

Reviewer 1 Report

Comments and Suggestions for Authors

see the attachment.

Comments on the Quality of English Language

minor edits are required. 

Reviewer 2 Report

Comments and Suggestions for Authors

This study explores collaborative target tracking by UAV swarms using a decentralized Kalman Filter with three different process models: Constant Velocity (CV), Constant Turn (CT), and Full State (FS). The target undergoes three phases of motion—linear, sinusoidal, and circular—allowing for a comprehensive assessment of the models' performance. Results indicate that the CV model excels in non-maneuvering scenarios, while the CT model is advantageous for maneuvering targets. The FS model, although providing a middle ground in performance, introduces higher computational complexity. The study concludes by emphasizing the impact of model selection on estimation and tracking tasks and suggests future directions for handling 3D motion and exploring interactions between different process models within the filter. 

Overall, the study contributes to the understanding of decentralized Kalman Filtering in UAV swarms, and minor refinements could improve its clarity and impact.

-       The methodology section is comprehensive, detailing the implementation of the decentralized Kalman Filter with different process models. Clarify any assumptions made and consider providing more context for readers unfamiliar with the models or algorithms.

-       Provide more interpretation regarding flocking behavior result(Figure 5). What is a reason for oscillating pattern? Any possible ideas on why FS provides better results that CT and CV?

-       The line interaction topology suggests that information is exchanged between neighboring UAVs in a sequential manner, possibly forming a chain-like communication structure. This topology choice can impact the efficiency and robustness of the swarm's collective behavior. What are limitations of line-based interaction concept? What are alternative approaches?

-       A more in-depth analysis of the implications of the findings and their significance in the broader context of swarm robotics and target tracking would strengthen the discussion section.

Comments on the Quality of English Language

The sentences are well-structured, and technical terminology is appropriately used. There are a few instances where sentence structures could be slightly refined for smoother readability. For example, in some parts, the use of complex sentences and technical details might make the text challenging for non-experts to follow. Consideration of a more straightforward presentation, especially in conveying complex concepts, could enhance overall clarity.

Reviewer 3 Report

Comments and Suggestions for Authors

1. The decentralized estimation and tracking of a mobile target using a swarm of UAVs is an innovative and forward-looking approach. This work showcases creativity in addressing real-world challenges.

2. The utilization of a flocking protocol for maintaining a collision-free formation is a commendable strategy. It reflects a thoughtful consideration of safety and efficiency in the UAV swarm's operation.

3. The thorough comparison of three different models for target motion (CV, CT, and FS) demonstrates a rigorous approach. This attention to detail contributes significantly to understanding the impact of model complexity on performance.

4. The use of Software-In-The-Loop (SITL) simulations in ROS/Gazebo is a strong point. It ensures a robust evaluation of the proposed models, enhancing the credibility of the study.

5. The conclusion provides clear insights into the implications of model selection on estimation and tracking tasks. The acknowledgment of future research directions, such as handling 3D movement and exploring interacting models, shows a forward-thinking perspective.

6. The work holds promise for real-world applications, especially with the consideration of additional dimensions like altitude. The exploration of consensus steps in the filter for merging benefits from multiple models indicates a practical mindset.

Overall, this research contributes to the field of decentralized UAV swarm technology and tracking systems. The findings and future directions outlined in the conclusion pave the way for continued advancements in the domain.
